

# Comparative mitogenomic analysis of mirid bugs (Hemiptera: Miridae) and evaluation of potential DNA barcoding markers

Juan Wang[*], Li Zhang[*], Qi-Lin Zhang[*], Min-Qiang Zhou, Xiao-Tong Wang, Xing-Zhuo Yang and Ming-Long Yuan

State Key Laboratory of Grassland Agro-Ecosystems, College of Pastoral Agricultural Science and Technology, Lanzhou University, Lanzhou, China

[*] These authors contributed equally to this work.

Corresponding author
Ming-Long Yuan, yuanml@lzu.edu.cn

## ABSTRACT

The family Miridae is one of the most species-rich families of insects. To better understand the diversity and evolution of mirids, we determined the mitogenome of *Lygus pratenszs* and re-sequenced the mitogenomes of four mirids (i.e., *Apolygus lucorum*, *Adelphocoris suturalis*, *Ade. fasciaticollis* and *Ade. lineolatus*). We performed a comparative analysis for 15 mitogenomic sequences representing 11 species of five genera within Miridae and evaluated the potential of these mitochondrial genes as molecular markers. Our results showed that the general mitogenomic features (gene content, gene arrangement, base composition and codon usage) were well conserved among these mirids. Four protein-coding genes (PCGs) (*cox1*, *cox3*, *nad1* and *nad3*) had no length variability, where *nad5* showed the largest size variation; no intraspecific length variation was found in PCGs. Two PCGs (*nad4* and *nad5*) showed relatively high substitution rates at the nucleotide and amino acid levels, where *cox1* had the lowest substitution rate. The Ka/Ks values for all PCGs were far lower than 1 (<0.59), but the Ka/Ks values of *cox1*-barcode sequences were always larger than 1 (1.34 – 15.20), indicating that the 658 bp sequences of *cox1* may be not the appropriate marker due to positive selection or selection relaxation. Phylogenetic analyses based on two concatenated mitogenomic datasets consistently supported the relationship of *Nesidiocoris* + (*Trigonotylus* + (*Adelphocoris* + (*Apolygus* + *Lygus*))), as revealed by *nad4*, *nad5*, *rrnL* and the combined 22 transfer RNA genes (tRNAs), respectively. Taken sequence length, substitution rate and phylogenetic signal together, the individual genes (*nad4*, *nad5* and *rrnL*) and the combined 22 tRNAs could been used as potential molecular markers for Miridae at various taxonomic levels. Our results suggest that it is essential to evaluate and select suitable markers for different taxa groups when performing phylogenetic, population genetic and species identification studies.

# INTRODUCTION

Mirid bugs (Hemiptera: Miridae) are one of the most species-rich families of insects, with approximately 11,000 described species in 1,200 genera (*Cassis & Schuh, 2012*; *Jung & Lee, 2012*). Mirid bugs play a key role in natural systems and agroecosystems, with a wide range of food preferences and behaviors (*Cassis & Schuh, 2012*; *Jung & Lee, 2012*; *Wheeler, 2001*). Some mirids are of great economic importance as pests on various cultivated plants (*Cassis & Schuh, 2012*; *Lu et al., 2010*), whereas others are beneficial species used as biological control agents (*Cassis & Schuh, 2012*). In China, several mirids (e.g., *Apolygus lucorum*, *Lygus pratenszs* and *Adelphocoris lineolatus*) are important insect pests on crops, vegetables and forages and recently have extensively increased population density on cotton due to increasing Bt cotton adoption (*Lu et al., 2010*). However, little is known about interspecific and intraspecific diversity and evolution in these mirids.

Mirids show high morphological diversity and some are difficult to be identified by eye due to small body size, especially closely related species with similar morphological characteristics. The mitochondrial cytochrome c oxidase subunit 1 (*cox1*) has been widely used as a molecular marker for molecular phylogenetics, population genetics and species identification in animals (*Avise, 2009*; *Hebert, Ratnasingham & Waard, 2003*; *Jinbo, Kato & Ito, 2011*; *Simon et al., 2006*). The effectiveness of *cox1* as a DNA barcoding marker has been widely investigated in many insect groups, such as Lepidoptera (*Cameron & Whiting, 2008*; *DeWaard et al., 2010*; *Wiemers & Fiedler, 2007*); Hemiptera (*Abd-Rabou et al., 2012*; *Foottit et al., 2008*; *Jung, Duwal & Lee, 2011*; *Park et al., 2011*; *Raupach et al., 2014*) and Coleoptera (*Kubisz et al., 2012*; *Monaghan et al., 2005*; *Raupach et al., 2010*). These studies showed that *cox1* was an effective and suitable DNA barcoding marker for most insect groups, but showed limited ability to identify closely related species for some groups (*Chi et al., 2012*; *Lee et al., 2013*; *Schmidt & Sperling, 2008*). Therefore, it is essential to explore other potential mitochondrial and nuclear markers for these groups, such as nuclear ITS (*Park et al., 2010*) and other mitochondrial genes (e.g., *nad4* and *nad5*) (*Brabec et al., 2015*; *Levkanicova & Bocak, 2009*; *Ye et al., 2015*; *Yu, Kong & Li, 2016*).

Insect mitogenome is a circular double-stranded molecule of 15–18 kb in size and usually contains 37 genes and a large non-coding region (known as control region) (*Boore, 1999*; *Cameron, 2014*). During the past decades, insect mitogenomes are the most extensively used genetic information for molecular evolutionary, phylogenetic and population genetic studies (*Cameron, 2014*; *Simon et al., 2006*). To date, only ten complete or nearly complete mitogenomes have been determined for Miridae. However, the number of sequenced mirid mitogenomes is still very limited compared to the species-richness of Miridae. Therefore, sequencing more mirid mitogenomes is essential for understanding the evolution of Miridae at the genomic level. In particular, all mirid mitogenomes available in GenBank were sequenced for just a single individual per species, which limited our understanding of intraspecific mitogenomic diversity. To date, knowledge about intraspecific evolution of insect mitogenomes is limited, with the notable exception of *Drosophila melanogaster*

**Table 1  List of mirid species analyzed in the study.**

| Subfamily | Species | Size (bp) | A+T% | AT-skew | GC-skew | GenBank accession | References |
|---|---|---|---|---|---|---|---|
| Bryocorinae | *Nesidiocoris tenuis* | 17,544 | 75.0 | 0.10 | −0.11 | NC_022677 | *Dai et al. (2012)* |
| Mirinae | *Adelphocoris fasciaticollis* | 15,434 | 77.4 | 0.16 | −0.22 | KJ001714 | *Wang et al. (2016a)*; *Wang et al. (2016b)* |
| | *Adelphocoris fasciaticollis_Yuan*[a] | 13,587 | 77.0 | 0.17 | −0.21 | KU234536 | This study |
| | *Adelphocoris lineolatus* | 15,595 | 77.1 | 0.16 | −0.21 | KJ020286 | *Wang et al. (2014b)* |
| | *Adelphocoris lineolatus_Yuan* | 15,433 | 76.9 | 0.16 | −0.21 | KU234537 | This study |
| | *Adelphocoris nigritylus*[a] | 14,522 | 77.2 | 0.17 | −0.21 | KJ020287 | *Wang et al. (2014b)* |
| | *Adelphocoris suturalis*[a] | 14,327 | 76.8 | 0.17 | −0.20 | KJ020288 | *Wang et al. (2014b)* |
| | *Adelphocoris suturalis_Yuan*[a] | 14,106 | 76.8 | 0.17 | −0.21 | KU234538 | This study |
| | *Apolygus lucorum* | 14,768 | 76.8 | 0.12 | −0.12 | NC_023083 | *Wang et al. (2014a)* |
| | *Apolygus lucorum_Yuan* | 15,647 | 76.8 | 0.11 | −0.12 | KU234539 | This study |
| | *Lygus hesperus* | 17,747 | 75.3 | 0.14 | −0.19 | NC_024641 | Unpublished |
| | *Lygus lineolaris* | 17,027 | 75.9 | 0.13 | −0.18 | NC_021975 | *Roehrdanz et al. (2016)* |
| | *Lygus pratenszs*[a] | 14,239 | 75.6 | 0.15 | −0.18 | KU234540 | This study |
| | *Lygus rugulipennis*[a] | 15,819 | 75.5 | 0.14 | −0.18 | KJ170898 | *Wang et al. (2014b)* |
| | *Trigonotylus caelestialium*[a] | 15,095 | 74.9 | 0.14 | −0.13 | KJ170899 | *Wang et al. (2014b)* |

**Notes.**
[a]Incomplete mitochondrial genomes.

(*Wolff et al., 2016*). Due to the linkage of these mitochondrial genes within the same mtDNA molecule, the same genealogy shared by mitochondrial genes is expected. However, incongruent phylogenetic results were frequently found among different mitochondrial genes (*Duchêne et al., 2011*; *Havird & Santos, 2014*; *Nadimi, Daubois & Hijri, 2016*). In addition, the single or a few concatenated genes could serve as a proxy for the entire mitogenomes (*Duchêne et al., 2011*; *Havird & Santos, 2014*; *Nadimi, Daubois & Hijri, 2016*), which provides a good opportunity to resolve phylogenetic relationships of Miridae that currently lacks sufficient entire mitogenome sequences. However, the performance of the best genes or regions is highly taxa-dependent (*Duchêne et al., 2011*; *Havird & Santos, 2014*; *Nadimi, Daubois & Hijri, 2016*). Therefore, it is needed to evaluate the potential and suitability of single mitochondrial genes as molecular markers within Miridae.

In this study, we sequenced and annotated the mitogenome of *L. pratenszs* and re-sequenced the mitogenomes of four mirid species (i.e., *Apo. lucorum*, *Ade. suturalis*, *Ade. fasciaticollis* and *Ade. lineolatus*). These five mirid bugs are important pests on crops, vegetables and forages in China (*Lu et al., 2010*; *Zhang et al., 2016*). Combined with ten mirid mitogenomes available from GenBank (Table 1), we provided a comparative mitogenomic analysis at various taxonomic levels. Particularly, we focused on molecular evolution of mitochondrial genes within genera and species. We also evaluated the potential of these mitochondrial genes as molecular markers by genetic distance and phylogenetic analyses. The results will provide useful genetic information for further

studies on phylogeny, species identification, phylogeography and population genetics in mirid bugs.

## MATERIALS AND METHODS

### Sample and DNA extraction

Adult specimens of five mirid bugs were collected from alfalfa field in Shishe Town, Xifeng District, Qingyang City, Gansu Province, China, in July 2013. Samples and voucher specimens have been deposited in the State Key Laboratory of Grassland Agro-Ecosystems, College of Pastoral Agricultural Science and Technology, Lanzhou University, Lanzhou, China. All specimens were initially preserved in 100% ethanol in the field, and transferred to −20 °C until used for DNA extraction. The total genomic DNA was extracted from thorax muscle of a single specimen using the Omega Insect DNA Kit (Omega Bio-Tek, Norcross, GA, USA) according to the manufacturer's protocols.

### PCR amplification and sequencing

For each mirid species, mitogenome was amplified with 10–13 overlapping fragments by using universal insect mitogenome primers (*Simon et al., 2006*) and species-specific primers designed from sequenced fragments. All primers used in this study are provided in Table S1 . PCR and sequence reactions were conducted following our previous studies (*Yuan et al., 2015a*; *Yuan et al., 2015b*).

### Genome annotation and sequence analysis

Sequence files were assembled into contigs with BioEdit 7.0.9.0 (*Hall, 1999*). Each of the five mirid mitogenomes newly sequenced in the present study was annotated following our previous studies (*Yuan et al., 2015a*; *Yuan et al., 2015b*). Nucleotide composition and codon usage were analyzed with MEGA 6.06 (*Tamura et al., 2013*). For each protein-coding gene (PCG) of all 15 mirid mitogenomes, the number of synonymous substitutions per synonymous site (Ks) and the number of nonsynonymous substitutions per nonsynonymous site (Ka) were calculated with MEGA 6.06 (*Tamura et al., 2013*). We also calculated the genetic distances for 13 PCGs and two ribosomal RNA genes (rRNAs) with MEGA 6.06 (*Tamura et al., 2013*) under the Kimura-2-parameter model (K2P). Strand asymmetry was calculated using the formulas: AT-skew = [A−T]/[A+T] and GC-skew = [G−C]/[G+C] (*Perna & Kocher, 1995*). To determine whether the Ka/Ks values of *cox1*-barcode sequences were relevant with the scope of sequences used, we downloaded all 7,759 *cox1* sequences of Miridae available in GenBank (March 1, 2017). After removing sequences shorter than 658 bp, a total of 2,326 sequences in 144 genera were obtained (Table S7). Except for forty genera with only one sequence, the remaining 104 genera were used to calculate the values of Ka, Ks and Ka/Ks.

### Phylogenetic analysis

Complete or nearly complete mitogenome sequences of eleven mirid bugs (15 samples, Table 1) and two outgroups from Pentatomomorpha (*Corizus tetraspilus* and *Eurydema gebleri*) (*Yuan et al., 2015a*; *Yuan et al., 2015b*) were used to perform phylogenetic analyses.

The complete sequences of 13 PCGs (excluding stop codons), two rRNAs and 22 transfer RNA genes (tRNAs) were used for phylogenetic analyses. Each PCG was aligned individually based on codon-based multiple alignments by using the MAFFT algorithm within the TranslatorX (*Abascal, Zardoya & Telford, 2010*) online platform. Sequences of each rRNA gene were individually aligned using the MAFFT v7.0 online server with G-INS-i strategy (*Katoh & Standley, 2013*). Alignments of individual genes were then concatenated as a combined matrix with DAMBE 5.3.74 (*Xia, 2013*). Two datasets were assembled for phylogenetic analyses: (1) nucleotide sequences of 13 PCGs (P123) with 11,133 residues and (2) nucleotide sequences of 13 PCGs, two rRNAs and 22 tRNAs (P123RT) with 14,905 residues. To evaluate phylogenetic potential of single mitochondrial genes, i.e., each of 13 PCGs, *rrnL* and *rrnS*, as well as the combined 22 tRNAs, were also used in phylogenetic analyses.

The optimal partitioning schemes and corresponding nucleotide substitution models for each datasets were determined by PartitionFinder v1.1.1 (*Lanfear et al., 2012*). We created input configuration files that contained pre-defined data blocks by genes and codons, e.g., 39 partitions for P123, 42 partitions for P123RT and 3 partitions for each of 13 PCGs. The "greedy" algorithm with branch lengths estimated as "unlinked" and Bayesian information criterion (BIC) were used to search for the best-fit scheme (Table S2). The best-fit partitioning schemes selected by PartitionFinder were used in all subsequent phylogenetic analyses. We used jModelTest 2.1.7 (*Posada, 2008*) to determine the best evolutionary model for *rrnL*, *rrnS* and the combined 22 tRNAs.

Phylogenetic analyses were conducted with Bayesian inference (BI) and maximum likelihood (ML) methods available on the CIPRES Science Gateway v3.3 (*Miller, Pfeiffer & Schwartz, 2010*). Bayesian analyses were performed with MrBayes 3.2.3 (*Ronquist et al., 2012*) on Extreme Science and Engineering Discovery Environment (XSEDE 8.0.24). Two independent runs with four chains (three heated and one cold) each were conducted simultaneously for $1 \times 10^6$ generations. Each run was sampled every 100 generations. Stationarity is assumed to be reached when ESS (estimated sample size) value is above 100 and PSRF (potential scale reduction factor) approach 1.0 as suggested in MrBayes 3.2.3 manual (*Ronquist et al., 2012*). The first 25% samples were discarded as burn-in, and the remaining trees were used to calculate posterior probabilities (PP) in a 50% majority-rule consensus tree. ML analyses were carried out using RAxML-HPC2 (*Stamatakis, 2014*) on XSEDE 8.0.24 with the GTRGAMMA model, and the node reliability was assessed by 1,000 bootstraps (BS).

## RESULTS

### General features of mirid mitogenomes

In the present study, we sequenced and annotated the mitogenomes of five mirid bugs: two were completely sequenced, whereas three were nearly complete mitogenomes (lacking sequences of three tRNAs and the putative control region) (Table 1, Table S3). The mitogenome sequences of five mirids have been deposited in GenBank of NCBI under

accession numbers: KU234536–KU234540. The two completely sequenced mitogenomes contained 37 typical mitochondrial genes (i.e., 13 PCGs, 22 tRNA genes and two rRNAs) and a large non-coding region (putative control region) (Table S3). The order and orientation of the mitochondrial genes were identical to that of the putative ancestral insect mitogenome (*Boore, 1999*; *Cameron, 2014*). Gene overlaps and spacers were presented in several conserved positions in the mirid mitogenomes, e.g., *trnS2/nad1* (7 bp), *trnW/trnC* (−8 bp), *atp8/atp6* (−7 bp) and *nad4/nad4L* (−7 bp).

The tRNAs in the three *Adelphocoris* species could be folded into a classical clover-leaf secondary structure (Fig. 1). However, *trnS1* (AGN) in *Apo. lucorum* and *L. pratenszs* species lacked the DHU stem-loop structures, as previously observed in many other true bugs (*Wang et al., 2014b*; *Yuan et al., 2015a*; *Yuan et al., 2015b*). All 22 tRNAs in *Apolygus* and *Lygus* species used the standard anticodon, whereas two tRNAs in the three *Adelphocoris* species were exceptions: *trnS1* was predicted to have anticodon UCU, whereas *trnK* had the anticodon UUU (Table S3). The sequences and structures of anticodon arms and aminoacyl acceptor stems were well conserved within Miridae, whereas most of the variations (nucleotide substitutions and indels) were found in the DHU loops, pseudouridine (TψC) arms and variable loops (Fig. 1).

All of the mirid mitogenomes showed similar nucleotide composition in the J-strand: high A+T content, positive AT- and negative GC-skews (Fig. S1), as is usually observed in insect mitogenomes (*Hassanin, Leger & Deutsch, 2005*). For each part of mitogenomes, the A+T content, AT- and GC-skews showed low variability among different taxonomic levels (i.e., family, subfamily, genus and species). For AT-skew, a negative value was found in PCGs, *rrnL*, *rrnS* and the 2nd codon position of PCGs, whereas the 1st and 3rd codon positions of PCGs had positive AT-skew values. Except for *Apo. lucorum*, a positive AT-skew value also was found for tRNAs in all mirids. For GC-skew, the 1st codon position of PCGs, *rrnL*, *rrnS* and tRNAs were markedly positive, whereas the 2nd and 3rd codon positions of PCGs showed negative values. The pattern of codon usage in all analyzed mirid mitogenomes was consistent with previous findings in insects (e.g., *Wang et al., 2015*; *Yuan et al., 2015a*), namely that A+T-rich codons were preferably used (Table S4).

## Gene variability in mirid mitogenomes

Among 13 PCGs, four genes (*cox1*, *cox3*, *nad1* and *nad3*) had no length variability in the 15 examined mirid mitogenomes (Table S5). Four genes showed size variation only in one species (i.e., *atp6*, *cox2* and *nad4* in *N. tenuis*, *cob* in *L. lineolaris*). *N. tenuis* belonging to Bryocorinae showed the most length differences with species from Mirinae. The length of *nad5* was most variable, but conserved within each genus. No length variation was found in the same species, whereas the most variations were found among genera. For the two genera including more than two species, no size variation was present in *Adelphocoris*, and only *cob* in *L. lineolaris* showed size difference with the others in *Lygus*. For *rrnL* and *rrnS*, intra-generic and -specific size differences were slight, whereas large differences were found among genera.

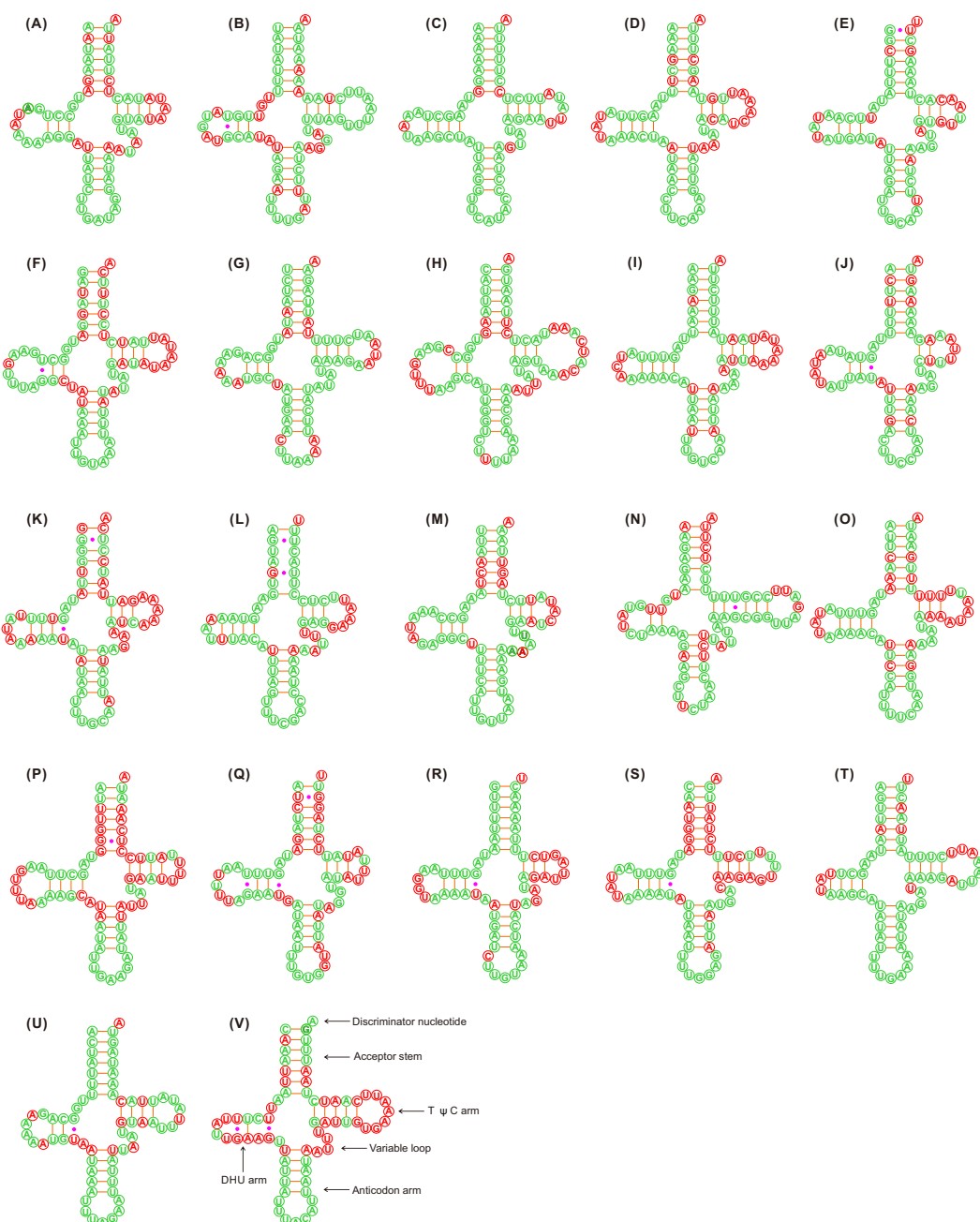

**Figure 1** **Putative secondary structures of the 22 tRNA genes identified in the mitochondrial genome of *Adelphocoris lineolatus*.** (A) *trnI*, (B) *trnQ*, (C) *trnM*, (D) *trnW*, (E) *trnC*, (F) *trnY*, (G) *trnL2*, (H) *trnK*, (I) *trnD*, (J) *trnG*, (K) *trnA*, (L) *trnR*, (M) *trnN*, (N) *trnS1*, (O) *trnE*, (P) *trnF*, (Q) *trnH*, (R) *trnT*, (S) *trnP*, (T) *trnS2*, (U) *trnL1*, (V) *trnV*. All tRNA genes are shown in the order of occurrence in the mitochondrial genome starting from *trnI*. The nucleotides showing 100% identity in the 15 mirid mitochondrial genomes are marked with green color, and the variable region are marked with red color. Bars indicate Watson-Crick base pairings, and dots between G and U pairs mark canonical base pairings in tRNA.

The numbers of variable and informative sites varied among genes and taxa (Table 2). The smallest gene *atp8* showed the highest informative sites in Miridae, Mirinae and Mirini, followed by *nad2* and *nad6*, while *nad4L* in *Adelphocoris* and *nad3* in *Lygus*. The *cox1* gene showed least informative sites in Miridae, Mirinae and Mirini, while *nad6* in *Adelphocoris* and *nad4L* in *Lygus*. Generally, *rrnL* and *rrnS* had the lower informative sites than most PCGs; *rrnL* was slightly higher than *rrnS*, except for *Adelphocoris*. Compared to the whole *cox1*, *cox1*-barcode sequences contained relatively small informative sites in most taxonomic levels (except for *Lygus*). Except for *cox1*, other two longest genes (*nad4* and *nad5*) contained moderate informative sites in most taxonomic levels (except for *Lygus*).

As expected, the largest genetic distances were found among subfamilies, followed by Tribes and genera, whereas the smallest among species (Fig. 2). Some genes (e.g., *atp6*, *cox2*, *nad1*, *nad3* and *nad6*) showed no intraspecific variations at nucleotide and/or amino acid levels, indicating that these genes were highly conserved. The values of K2P and Ka in 13 PCGs of *Lygus* were larger than that of *Adelphocoris*, indicating that *Lygus* had the higher substitution rates. For the K2P distance, *nad6* was the highest in Miridae and *Lygus*, whereas *atp8* in Mirinae and Mirini. The K2P distance of *nad4L* was the highest within *Adelphocoris*, and relatively high K2P distance was also found in Miridae, Mirinae, Mirini and *Lygus*. The two rRNAs showed similar K2P distance with those of *cox1-3* and *cob* in Miridae, Mirinae and Mirini, but had the lowest substitution rates in *Lygus*. For Miridae, Mirinae and Mirini, *atp8* showed the highest Ka value, whereas *nad4L* in *Adelphocoris* and *nad6* in *Lygus* showed the highest Ka values. Two genes (*nad4* and *nad5*) showed relatively high substitution rates at the nucleotide and amino acid levels. *Cox1* showed the lowest substitution rate in Miridae, Mirinae, Mirini, *Adelphocoris* and *Lygus*. The Ka/Ks values for all PCGs were far lower than 1 (<0.59) (Fig. 2), suggesting that these genes were evolving under purifying selection. However, we found that the Ka/Ks values for *cox1*-barcode sequences of the 15 mirids were always larger than 1 (1.34 in *Ade. suturalis* to 15.20 in Mirini; Table S6). In addition, analyses for *cox1*-barcode sequences of Miridae from GenBank showed that forty-four genera had no Ka and/or Ks values (Table S7), whereas the values of Ka/Ks for the remaining 61 genera were larger than 1 (1.5–263.0, Table S7), indicating that these sequences may be under positive selection or selection relaxation during the evolutionary process of Miridae. These results suggested that different genes had different substitution rates, whereas the same genes in different taxonomic levels showed large differences.

## Mirid phylogeny based on combined mitochondrial genes

Phylogenetic relationships of 15 mirid mitogenome sequences were inferred using BI and ML methods based on two mitogenomic datasets (P123 and P123RT). The results showed that the two methods with the same dataset resulted in identical tree topology, whereas slight difference was found in the relationships of three species between the two datasets (Fig. 3). For the P123 dataset, *Ade. nigritylus* was sister to *Ade. suturalis* (PP = 0.65, BS = 84). For the P123RT dataset, *Ade. nigritylus* had a closer relationship with *Ade. fasciaticolli* and *Ade. lineolatus* (PP = 0.64, BS = 30), which was consistent with previous mirid phylogeny based on mitogenomic data (*Wang et al., 2014b*). However, low supports

Wang et al. (2017), *PeerJ*, DOI 10.7717/peerj.3661

**Table 2  Number of variable sites and number of informative sites in Miridae.**

| Gene | Miridae | | Mirinae | | Mirini | | *Adelphocoris* | | *Lygus* | |
|---|---|---|---|---|---|---|---|---|---|---|
| | Number of variable sites (%) | Number of informative sites (%) | Number of variable sites (%) | Number of informative sites (%) | Number of variable sites (%) | Number of informative sites (%) | Number of variable sites (%) | Number of informative sites (%) | Number of variable sites (%) | Number of informative sites (%) |
| *atp6* | 319 (47.47) | 212 (31.55) | 260 (38.86) | 203 (30.34) | 227 (33.93) | 199 (29.75) | 39 (5.83) | 23 (3.44) | 50 (7.47) | 9 (1.35) |
| *atp8* | 84 (52.83) | 60 (37.74) | 74 (46.54) | 56 (35.22) | 58 (36.48) | 56 (35.22) | 4 (2.56) | 3 (1.92) | 7 (4.40) | 2 (1.26) |
| *cob* | 412 (36.43) | 273 (24.14) | 350 (30.95) | 263 (23.25) | 296 (26.17) | 254 (22.46) | 51 (4.51) | 39 (3.45) | 117 (10.34) | 19 (1.68) |
| *cox1* | 534 (34.83) | 344 (22.44) | 425 (27.72) | 322 (21.00) | 357 (23.29) | 314 (20.48) | 87 (5.68) | 60 (3.91) | 110 (7.18) | 23 (1.50) |
| *cox2* | 267 (39.38) | 185 (27.29) | 219 (32.30) | 168 (24.78) | 183 (26.99) | 164 (24.19) | 37 (5.46) | 23 (3.39) | 53 (7.82) | 10 (1.47) |
| *cox3* | 311 (39.72) | 204 (26.05) | 252 (32.18) | 191 (24.39) | 212 (27.08) | 189 (24.14) | 36 (4.06) | 25 (3.19) | 62 (7.92) | 11 (1.40) |
| *nad1* | 343 (37.12) | 215 (23.27) | 274 (29.65) | 194 (21.00) | 215 (23.27) | 190 (20.56) | 27 (2.92) | 19 (2.06) | 75 (8.12) | 12 (1.30) |
| *nad2* | 562 (55.92) | 352 (35.02) | 441 (43.88) | 324 (32.24) | 363 (36.12) | 316 (31.44) | 67 (6.75) | 32 (3.22) | 91 (9.05) | 11 (1.09) |
| *nad3* | 168 (47.86) | 103 (29.34) | 132 (37.61) | 96 (27.35) | 109 (31.05) | 94 (26.78) | 14 (3.99) | 10 (2.85) | 32 (9.12) | 7 (1.99) |
| *nad4* | 703 (52.96) | 450 (33.86) | 570 (42.99) | 404 (30.47) | 452 (34.09) | 399 (30.09) | 74 (5.58) | 46 (3.47) | 112 (8.45) | 23 (1.73) |
| *nad4L* | 167 (54.58) | 101 (33.01) | 130 (42.48) | 94 (30.72) | 105 (34.31) | 94 (30.72) | 20 (6.60) | 13 (4.29) | 30 (9.90) | 3 (0.99) |
| *nad5* | 828 (48.51) | 541 (31.69) | 678 (39.72) | 506 (29.64) | 550 (32.22) | 496 (29.06) | 69 (4.07) | 56 (3.30) | 120 (7.07) | 19 (1.12) |
| *nad6* | 289 (57.68) | 172 (34.33) | 228 (46.63) | 155 (31.70) | 172 (35.17) | 151 (30.88) | 18 (3.68) | 9 (1.84) | 51 (10.49) | 7 (1.44) |
| *rrnL* | 501 (39.60) | 321 (25.38) | 388 (30.77) | 285 (22.60) | 301 (23.91) | 283 (22.48) | 28 (2.27) | 17 (1.38) | 47 (3.75) | 9 (0.72) |
| *rrnS* | 494 (55.01) | 211 (23.50) | 205 (23.06) | 198 (22.27) | 205 (23.16) | 193 (21.81) | 36 (4.47) | 12 (1.49) | 34 (3.99) | 4 (0.47) |
| *cox1*-barcoding sequences | 221 (33.59) | 135 (20.52) | 176 (26.75) | 125 (19.00) | 142 (21.58) | 121 (18.39) | 28 (4.26) | 24 (3.65) | 48 (7.29) | 13 (1.98) |

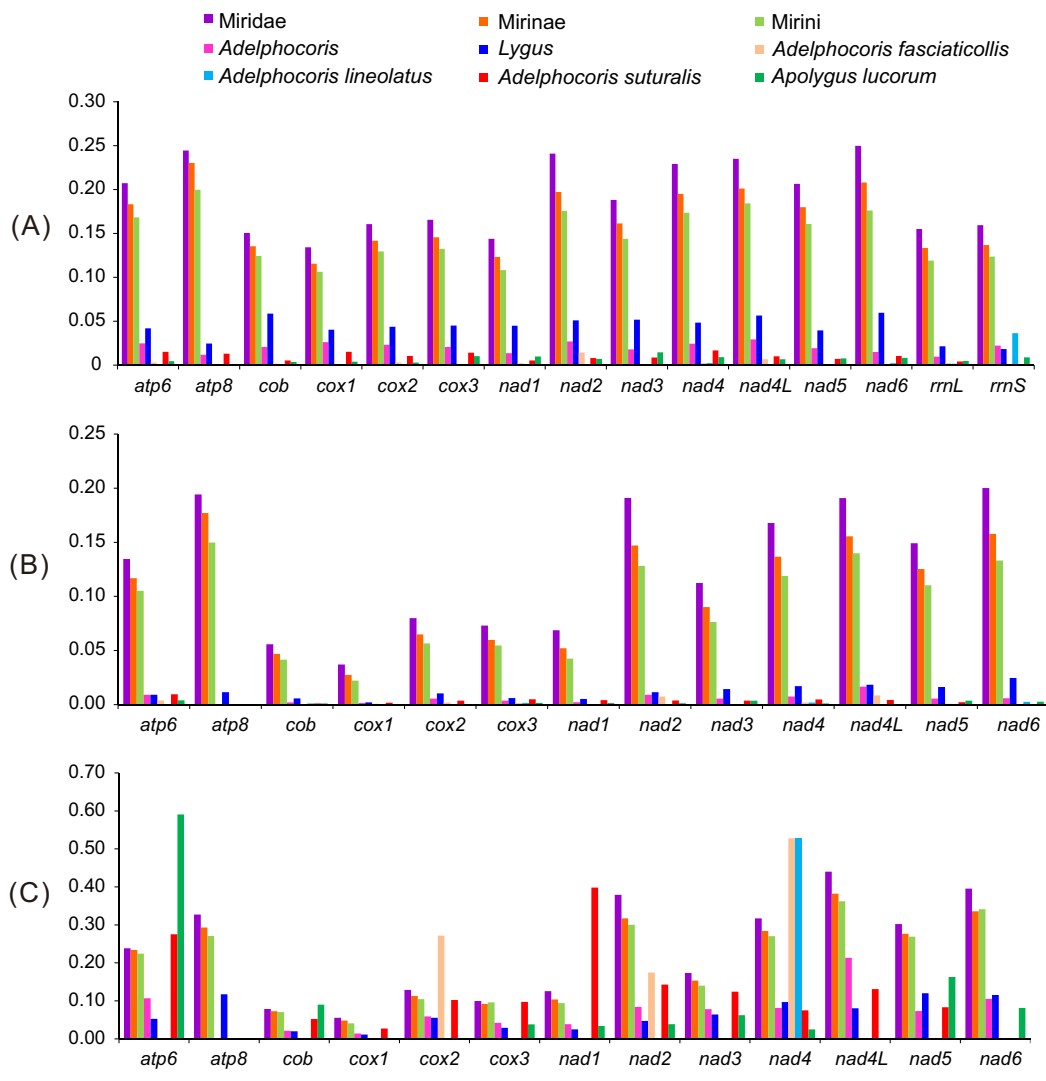

**Figure 2** **The K2P genetic distance, Ka and Ka/Ks of 13 protein-coding genes among the 15 mirid mitochondrial genomes.** (A) K2P, the Kimura-2- parameter distance; (B) Ka, the number of nonsynonymous substitutions per nonsynonymous site; (C) Ka/Ks. Ks, the number of synonymous substitutions per synonymous site.

were present in both analyses (Fig. 3), indicating that the phylogenetic position of *Ade. nigritylus* was unstable. All analyses consistently supported the relationship of *Nesidiocoris* + (*Trigonotylus* + (*Adelphocoris* + (*Apolygus* + *Lygus*))), as previous mitogenomic analyses (*Wang et al., 2014b*). For *Lygus*, the two datasets consistently recovered a phylogeny of (*rugulipennis* + (*lineolaris* + (*hesperus* + *pratenszs*))). The monophyly of *Adelphocoris* was strongly supported by all analyses with high supports (PP = 1.0, BS = 100), as was the monophyly of *Lygus* (PP = 1.0, BS = 100).

## Mirid phylogeny based on single mitochondrial gene
We performed phylogenetic analyses using BI and ML methods with single mitochondrial genes, including each of 13 PCGs, *rrnL*, *rrnS* and the combined 22 tRNAs (Figs. S2

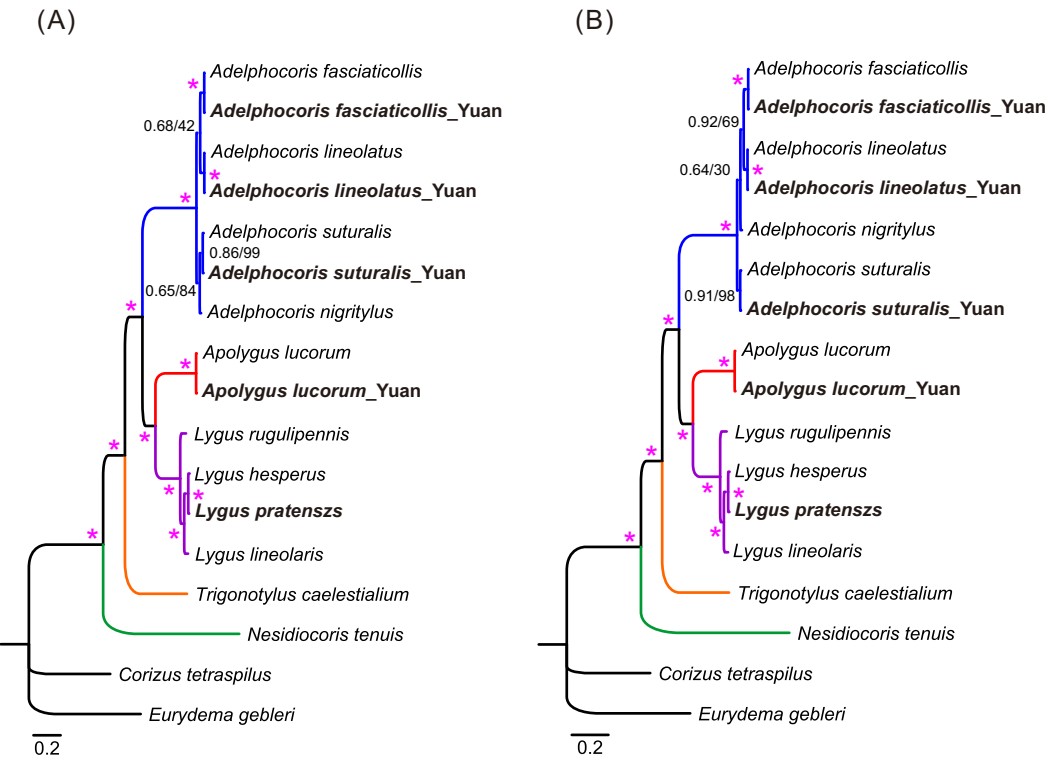

**Figure 3** **The mitochondrial phylogeny of eleven mirid bugs based on the two combined datasets: (A) P123 and (B) P123RT.** Numbers on branches are Bayesian posterior probabilities (PP, before slash) and Bootstrap values (BS, after slash). Asterisk (*) indicates PP = 1.0 and BS = 100. Species sequenced in the present study are bold.

and S3). The results showed that the tree topologies were variable among different datasets, indicating incongruent phylogenetic signals among genes, as reported in previous similar studies (*Duchêne et al., 2011*; *Havird & Santos, 2014*; *Nadimi, Daubois & Hijri, 2016*). However, all analyses consistently supported the monophyly of each of *Adelphocoris* (PP = 0.87–1.0, BS = 82–100) and *Lygus* (PP = 0.74–1.0, BS = 59–100), and several relationships within each of these two genera were recovered by different individual genes (Table 3, Figs. S2 and S3). Four datasets (*nad4*, *nad5*, *rrnL* and tRNAs) with the two analytical methods supported the phylogeny: *Nesidiocoris* + (*Trigonotylus* + (*Adelphocoris* + (*Apolygus* + *Lygus*))), as recovered by the P123 and P123RT datasets. This relationship among the five genera was also supported by BI analyses with two other genes (*cox1* and *nad2*). These four datasets supported the *Lygus* phylogeny of (*rugulipennis* + (*lineolaris* + (*hesperus* + *pratenszs*))), as four other genes (*cob*, *cox3*, *nad1* and *nad6*). For *Adelphocoris*, BI and ML analyses of four genes (*cox2*, *cox3*, *nad5* and *rrnL*) supported (*fasciaticollis* + *lineolatus*) + *nigritylus* (PP = 0.85–1.0, BS = 34–98), whereas other four genes (*atp6*, *cox1*, *nad1* and *nad4*) recovered the sister relationship of *nigritylus* and *suturalis* (PP = 0.63–0.9, BS = 53–97). It was notable that the *cox1*-barcode sequences did not support the sister relationship of *Apolygus* + *Lygus* consistently recovered by many independent datasets (Fig. 3 and Figs. S2–S4).

**Table 3** The phylogeny for the major clades of Miridae recovered by different mitochondrial datasets and analytical approaches.

| Gene | *Adelphocoris* | *Lygus* | Nes+(Tri+ (Ade+(Apo+Lys))) | Af+Al | (Af+Al)+An | An+As | Lr+(Ll+ (Lh+Lp)) |
|---|---|---|---|---|---|---|---|
| *atp6* | M/M | M/M | N/N | N/N | N/N | Y/Y | Y/N |
| *atp8* | M/M | M/M | N/N | N/N | N/N | N/N | N/N |
| *cob* | M/M | M/M | N/N | Y/Y | N/N | N/N | Y/Y |
| *cox1* | M/M | M/M | Y/N | N/Y | N/N | Y/Y | N/Y |
| *cox2* | M/M | M/M | N/N | Y/Y | Y/Y | N/N | Y/N |
| *cox3* | M/M | M/M | N/N | Y/Y | Y/Y | N/N | Y/Y |
| *nad1* | M/M | M/M | N/N | N/N | N/N | Y/Y | Y/Y |
| *nad2* | M/M | M/M | Y/N | N/N | N/N | N/N | N/N |
| *nad3* | M/M | M/M | N/N | N/Y | N/N | N/N | N/N |
| *nad4* | M/M | M/M | Y/Y | Y/Y | N/N | Y/Y | Y/Y |
| *nad4L* | M/M | M/P | N/N | N/N | N/N | Y/N | Y/N |
| *nad5* | M/M | M/M | Y/Y | Y/Y | Y/Y | N/N | Y/Y |
| *nad6* | M/M | M/M | N/N | N/N | N/N | N/N | Y/Y |
| *rrnL* | M/M | M/M | Y/Y | Y/Y | Y/Y | N/N | Y/Y |
| *rrnS* | M/M | M/M | N/N | N/N | N/N | N/N | N/N |
| 22 tRNAs | M/M | M/M | Y/Y | Y/Y | N/N | N/Y | Y/Y |
| P123 | M/M | M/M | Y/Y | Y/Y | N/N | Y/Y | Y/Y |
| P123RT | M/M | M/M | Y/Y | Y/Y | Y/Y | N/N | Y/Y |
| *cox1*-barcoding sequences | M/M | M/M | N/N | Y/Y | N/N | Y/Y | Y/Y |

**Notes.**

Results from left to right are obtained from Bayesian inference and maximum likelihood, respectively.

M, monophyletic; P, paraphyletic or polyphyletic; Y, yes a phylogeny is supported; N, no a phylogeny is not supported.; Nes, *Nesidiocoris*; Tri, *Trigonotylus*; Ade, *Adelphocoris*; Apo, *Apolygus*; Lys, *Lygus*; Af, *Adelphocoris fasciaticollis*; Al, *Ade. lineolatus*; An, *Ade. nigritylus*; As, *Ade. suturalis*; Lr, *Lygus ruguli­pennis*; Ll, *L. lineolaris*; Lh, *L. hesperus*; Lp, *L. pratenszs*.

## DISCUSSION

To date, a total of 15 mitogenomes representing 11 species in five genera were sequenced for Miridae. For the seven nearly complete mitogenomes, the undetermined region was the control region characterized by notable base composition bias, high numbers of tandem repeats and stable stem-loop structures. These features may result in disruption of PCR and sequencing reactions, as has been reported in other true bugs (*Wang et al., 2014b*; *Yuan et al., 2015a*). The size of completely sequenced mitogenomes greatly varied among genera, ranging from 14,522 bp in *Ade. nigritylus* and 17,747 bp in *L. hesperus*, primarily due to the significant size variation of the control region. The complete mitogenome of *Apo. lucorum* (15,647 bp) re-sequenced in the present study was rather larger than that previously sequenced by *Wang et al. (2014a)* (14,768 bp), largely due to significant size difference between the two control regions. Sequence alignment of control regions found that *Apo. lucorum* previously sequenced lacked regions of multiple repeated sequences, implying that the control region may be incomplete. The two completely sequenced mitogenomes of *Ade. lineolatus* were highly similar in length, with a 162 bp difference. Generally, genome size and total length of spacers and overlaps were more conserved within genus and species

than that within subfamily and family, as reported in previous similar studies (*Roehrdanz et al., 2016*; *Wang et al., 2014a*; *Wang et al., 2014b*; *Wang et al., 2016a*).

The loss of the DHU arm in *trnS1* (AGN) has been considered a typical feature of insect mitogenomes (*Cameron, 2014*). It has been shown that in the nematode *Ascaris suum* the tRNA genes that lack the DHU arm are functional (*Okimoto et al., 1992*). We found that *trnS1* in *Apo. lucorum* and *L. pratenszs* species lacked the DHU arm, but this tRNA in all sequenced mitogenomes of *Adelphocoris* species had a classical clover-leaf secondary structure, indicating the diversity in the secondary structures of *trnS1* within miridae. With the exception of *trnS1* and *trnK*, all the remaining 20 tRNAs used the same anticodon in mirids as in other hemipterans (*Wang et al., 2015*; *Yuan et al., 2015a*; *Yuan et al., 2015b*). In the four *Adelphocoris* species, *trnS1* changed the anticodon GCU with UCU and *trnK* changed the anticodon CUU with UUU. We re-sequenced the mitogenomes of three *Adelphocoris* species, confirming that the two anticodons of *trnS1* and *trnK* were genus-specific conserved. Therefore, the variations in structures and anticodons of mirid tRNAs may be genus-specific, and may indicate the high species diversity of Miridae. The mutations in the *trnS1* and *trnK* anticodons were uncommon in hemipteran mitogenomes, which may be correlated with the AGG codon reassignments between serine and lysine (*Abascal et al., 2006*; *Wang et al., 2014b*). We also noticed that *trnS1* in coleopteran mitogenomes always used the anticodon UCU (*Sheffield et al., 2008*; *Yuan et al., 2016*) and *trnK* used UUU in some beetles (*Li et al., 2016*; *Wang et al., 2016b*), suggesting the parallel evolution of AGG codon reassignments and point mutations at the anticodons of *trnS1*/*trnK* in insect mitogenomes (*Abascal et al., 2006*; *Wang et al., 2014b*). Further study by sequencing more mitogenomes from other genera and species is needed to investigate the evolution of anticodons and structures within Miridae.

Currently, *cox1* has been extensively used as DNA barcoding for evaluating and resolving phylogenetic relationships in insects (*Hebert, Ratnasingham & Waard, 2003*; *Jinbo, Kato & Ito, 2011*). Although the whole *cox1* sequences were evolving under purifying selection, the Ka/Ks values of the *cox1*-barcode sequences were always larger than 1 at various taxonomic levels within Miridae. Therefore, when we aim to determine the neutral population structure of mirids, the whole *cox1* sequences, combined other mitochondrial genes (e.g., *nad4* and *nad5*) as well as other markers (e.g., microsatellites and SNPs), may be preferred. In contrast, the *cox1*-barcode sequences may have the potential to study the adaptive evolution of mirids in the future. However, further investigations with denser sampling and additional analytical methods for the Ka/Ks ratio (e.g., Bayesian methods) are essential to reveal evolutionary patterns of *cox1*-barcode sequences within Miridae.

Phylogenetic analyses indicated that individual genes supported different phylogenetic relationships of the 15 mirids despite their linked nature, as reported in previous studies (*Duchêne et al., 2011*; *Havird & Santos, 2014*; *Nadimi, Daubois & Hijri, 2016*; *Seixas, Paiva & Russo, 2016*). The most probable factor for these incongruent phylogenies is the lack of adequate phylogenetic information within each individual gene, but other potential factors (e.g., phylogenetic inference methods, incomplete taxon sampling) should be considered. Two PCGs (*nad4* and *nad5*) consistently supported the same relationships

among the five genera and the four *Lygus* species, as the two concatenated mitogenomic datasets. These two genes were also identified as good molecular markers for metazoan phylogenetic analyses (*Havird & Santos, 2014*). However, most previous studies focused on the phylogenetic performance of mitochondrial PCGs, whereas the phylogenetic potential of RNA genes were rarely assessed by comparing the performance of single and concatenated mitochondrial genes. In the present study, we found that *rrnL* and the combined 22 tRNAs performed well in phylogenetic analyses, as *nad4* and *nad5*, suggesting their potential and importance in resolving the phylogeny of Miridae. However, we found that although *cox1* probably was the most commonly used mitochondrial genes in studies of metazoans, this gene (especially *cox1*-barcode sequences) showed poor phylogenetic performance within Miridae, indicating that the *cox1* sequences provided phylogenetic signals that may not be representative of the other mitochondrial genes in Miridae. Previous studies showed incongruent phylogenetic results of *cox1* sequences, suggesting that the suitability of *cox1* may be taxa-specific and relevant to species number used (*Havird & Santos, 2014*). Therefore, we should be cautious when the phylogenies are solely derived from sequence data of *cox1*, whereas additional mitochondrial genes could provide useful genetic information for phylogenetics and population genetics studies of mirid bugs.

## CONCLUSIONS

In this study, we determined the mitogenomes of *L. pratenszs* and re-sequenced other four mirid mitogenomes, and provided a comparative analysis for all 15 sequenced mitogenomes at various taxonomic levels. The results showed that gene content, gene arrangement, codon usage and nucleotide composition were well conserved within Miridae. Four protein-coding genes (*cox1*, *cox3*, *nad1* and *nad3*) had no length variability, where *nad5* showed the most size variation; no intraspecific length variation was found in PCGs. Two genes (*nad4* and *nad5*) showed relatively high substitution rates and informative sites in most taxonomic levels, where *cox1* had the lowest in Miridae, Mirinae and Mirini. Taken sequence length, substitution rate and phylogenetic signal together, the individual genes (*nad4*, *nad5* and *rrnL*) and the combined dataset of 22 tRNAs could be used as potential molecular markers for Miridae. Our results suggest that it is essential to evaluate and select suitable markers for different taxa groups when performing phylogenetic and population genetic studies.

### Funding

The study was funded by the Natural Science Foundation of Gansu Province (1506RJZA211), the Fundamental Research Funds for the Central Universities (lzujbky-2016-5) and the National Key Technology Support Program (2014B AD14B006). The funders had no role in study design, data collection and analysis, decision to publish, or preparation of the manuscript.

## Grant Disclosures

The following grant information was disclosed by the authors:

Natural Science Foundation of Gansu Province: 1506RJZA211.

Fundamental Research Funds for the Central Universities: lzujbky-2016-5.

National Key Technology Support Program: 2014B AD14B006.

## Competing Interests

The authors declare there are no competing interests.

## Author Contributions

- Juan Wang performed the experiments, analyzed the data, wrote the paper, prepared figures and/or tables, collected the insect samples.
- Li Zhang performed the experiments, analyzed the data, wrote the paper, prepared figures and/or tables.
- Qi-Lin Zhang performed the experiments, analyzed the data, prepared figures and/or tables, reviewed drafts of the paper, collected the insect samples.
- Min-Qiang Zhou, Xiao-Tong Wang and Xing-Zhuo Yang performed the experiments.
- Ming-Long Yuan conceived and designed the experiments, analyzed the data, contributed reagents/materials/analysis tools, wrote the paper, prepared figures and/or tables, reviewed drafts of the paper, collected the insect samples.

## DNA Deposition

The following information was supplied regarding the deposition of DNA sequences:

All mitochondrial genomes here are accessible via GenBank accession numbers KU234536–KU234540.

## Data Availability

The raw data has been supplied as Supplementary Files.

## Supplemental Information

Supplemental information for this article can be found online at http://dx.doi.org/10.7717/peerj.3661#supplemental-information.

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
