# Peer review of "Comparative mitogenomic analysis of mirid bugs (Hemiptera: Miridae) and evaluation of potential DNA barcoding markers"

_PeerJ, doi:10.7717/peerj.3661_

## Round 0.1 · original submission · Major Revisions

Thank you very much for submitting your manuscript to PeerJ. Your manuscript was fully evaluated at the editorial level and by independent peer reviewers. The reviewers appreciated the insights into phylomitogenomics and evolution of an understudied group of insect. However, they also identified some aspects of the manuscript that could be improved.

Thus, I would like to ask you to reply to all of the reviewers’ comments and identify the points below as crucial for the revision:

1) Justification of the choice of these particular species for sequencing and re-sequencing, as requested by both reviewers.
2) Further investigation on cox-1 gene, as suggested by both reviewers.
3) Separating results, discussion and conclusions, as a means to improve readability of the manuscript, as suggested by reviewer 2.
4) Revision by a native English speaker, to make the text easier to follow.
5) Providing additional information, clarifications and discussion requested by the reviewers.

Reviewer 1 ·

Basic reporting

In the study (ID: 14083) entitled “Comparative mitogenomic analysis of mirid bugs (Hemiptera: Miridae) and evaluation of potential DNA barcoding markers,” the authors conduct a comparative study of the mitochondrial genomes of 11 mirid species. Five genomes (Apolygus lucorum, Adelphocoris suturalis, Ade. fasciaticollis, Ade. lineolatus and Lygus pratensis) were sequenced for the particular study while the remaining were retrieved from previously published work (I assume from NCBI even though this is not explicitly mentioned). The authors, measure and discuss the length variability, the genetic distance, the ka/ks ratio, the nucleotide composition, codon usage, and the phylogenetic signal of different genes as well as their order and orientation.

For the gene annotation and sequence analyses, the authors followed the steps described in related previous publications (Yuan et al., 2015a,b) of the group. For the phylogenetic analyses, they used standard software such as RAxML and MrBayes.

Experimental design

Please find below my major concerns regarding the study:

1. The Miridae family, as noted by the authors, is a particularly species-rich group with many of its species being of economic interest given their pest-behavior. For such a diverse and largely unknown taxon, any insight to its genomic content and evolution is a significant contribution. However, the framework and working hypothesis in the particular study is not clearly described in the manuscript. The authors should explain to some extent why they chose to sequence the specific taxa, especially since the same ones have been sequenced before in Wang et al. (2014). A potential motivation could have been to explore the intraspecific mitogenomic variation (briefly implied in the introduction, lines 68-70). However, this rationale is not identified in the results/discussion section.

2. The paper suffers from syntax problems (e.g. lines 68-70, 73-75, 165-167, 169-171, 192-194, 211-212), and lack of coherence, which makes it hard to follow. Below are some examples that highlight some problems of the manuscript:

Line 109. It is not clear whether the analyses mentioned in “Genome annotation and sequence analysis” have been conducted for the five newly sequenced samples or all mitogenomes involved in the study.

Lines 120 -122. The section “Phylogenetic analysis” starts with the sentence “Complete or nearly complete mitogenome sequences of eleven mirid bugs and two outgroups.” However, in Figure 2 there are 11 mirid species but 15 samples. The authors should describe much clearer what and how many sequences they used in their analyses.

There is no information in the manuscript, information about the sequences (e.g. which species) retrieved from previous publications. Some information can only be retrieved by the figures of the paper or by referring to previous related publications (i.e. Wang et al., 2014).

Several acronyms are used in the manuscript without defining them in their first occurrence [e.g. PCG (mentioned in abstract, not in main text), K2P, XSEDE]

The figure and table legends need to be explained in more detail.

Validity of the findings

3. The authors find that the COI sequences are characterized by ka/ks ratio, higher than 1, based on these findings they argue that COI might not be eligible for phylogenetic analyses. It seems that the ka/ks ratio changes widely depending on the sequences they use (i.e., 1.34 – 15.20). In NCBI there are more than 7000 COI entries. Therefore, it would be worth to cross check these findings based on more of the available data. Additionally, given the extensive use of COI as a barcoding gene in Miridae it would be important to discuss further what are the expected implications of using a marker under positive selection.

Reviewer 2 ·

Basic reporting

This is an interesting paper by Wang and colleagues, on comparative mitogenomic analysis of mirid bugs and evaluation of potential DNA barcoding markers. However, there are parts of the manuscript (ms) where English used are not clear enough and unambiguous that sometimes make it hard for the reader to follow and demands a lot of back and forth reading of different sections of the paper. I believe that this could be significantly improved if a native English speaking colleague will review the manuscript.
The introduction and the provided background are sufficient to demonstrate how the work fits into the broader field of knowledge. In general, the structure of the manuscript conforms to the instructions for authors provided by the journal. However, a clear Discussion section is lacking despite explanations and justifications provided in the Results section of the ms. I suggest a rearrangement in the Results and Conclusions sections in such a way as to also have a Discussion section. This will significantly improve the flow of the ms for the reader and communicate the results of the present study more easily and presumably to a greater audience.

Experimental design

The research is within the aims and scope of the journal. However, providing more details on the reasoning of species selection as well as on the significance of species identification of these particular species (e.g. and not other pests) will greatly improve the ms. The investigation conducted is rigorous and has been performed to a high technical standard. Furthermore, methods employed have been described with sufficient detail and information.

Validity of the findings

The data on which conclusions are based are robust. At the results section and specifically the part referring to the tRNAs, a more complete image could greatly benefit the reader i.e. comparison of tRNAs in all studied mitogenomes should be mentioned.

Additional comments

This is a very interesting study that in my opinion could further improve from the proposed minor changes. Throughout the ms, although you are referring to “species delimitation” it is evident that you mean “species identification” since cox1 is the marker used for DNA barcoding. Please make the appropriate changes throughout the ms.
In general, and when referring a) to already existing sequences and b) samples sequenced in the present study as well as on their characteristics it is rather difficult and confusing for the reader to understand to what you are referring to each time. Please make the appropriate changes throughout the ms in order to be clearer.
Regarding the Ka/Ks ratio estimation of the barcoding part of cox1 gene I believe that the analysis is not convincing enough. Taking only part of a gene and testing for selection is somehow inaccurate since in order to support the results and the conclusions drawn based on these. A more admissible solution would be to test selection in similarly sized subunits although entire gene sequences are always preferable.
I also have some comments and suggestions for particular lines in the ms:
Introduction
L59 Please give examples of mtDNA genes used in the references that you provide (as you did for ITS)
L61 please replace “double-strand molecule” with “double-stranded molecule”
L64 please replace with “molecular evolutionary, phylogenetic and population genetic studies”
L68-L70 please rephrase since it is not clear whether a single individual was sequenced per species or all sequenced mitogenomes where from individuals of the same species despite that it becomes evident later on in the ms
L70 please explain to what “these” is referring to
L71 sharing the same underlying genealogy does not mean that different genes will have the same phylogenetic signal. Please replace “identical phylogenetic signal” with “share the same genealogy” or something equivalent
L73-L77 please rephrase and give more information on the “lack of sufficient mitogenome sequences”
M & M
L105 please replace “were” with “are”
L140-L141 please explain why you also used jModeltest
L148 please rephrase to “as suggested in MrBayes 3.2.3 manual”
Results
L169 please replace to “Gene overlaps and spacer were present in several…”
L183 in Figure S1 AT and GC skewness are the opposite than those described in text also see L190-L192
L194 please rephrase to “were preferably used”
L201 “No length was found” do you mean length variation?
L259 please explain what you mean by “some relationships”
L269-L272 Cox1-barcode sequences are usually employed for species identification and not in phylogenetic studies where the complete or nearly complete sequence of the gene or better multiple genes are preferable.
Table3
L491 please explain what you mean by a feature is (or is not) present
Figure 2
L455-L458 please indicate at the figure’s legend what asterisks (*) stand for

---

## Round 0.2 · Minor Revisions

Thank you very much for the thoughtful revision of the manuscript. After careful consideration, I have concluded that at this point your manuscript requires an additional minor revision. I would like to ask you to include few additional changes based on the comments made by the two experts, especially:

- Discuss the Ka/Ks ratios obtained in the light of the existing literature, with the possible explanations of the differences, as suggested by the Reviewer 1.
- Address the comment on “evolutionary rates of mitochondrial DNA”, as suggested by the Reviewer2.

I am sorry to hear that there was a problem with legends for Figures 1 to 3 in the version of the Reviewer 2. This figure legends are present and seem complete in my version of the revised manuscript, and I have contacted technical stuff to check what the problem was.

Finally, I have spotted two minor mistakes (numbering based on the pdf file):
Line 183 – “conserved”, not “conversed”
Line 3611 – “be” not “been”

Reviewer 1 ·

Basic reporting

In the current version of the manuscript the authors are describing five mirid mitochondrial genomes and compare them with previously published genomes of the same or related species in order to identify the genes that may be useful in a phylogenetic framework.
In their analysis and results they place great emphasis on the Ka/Ks ratio across the different coding genes and the amount of congruence among independent gene-tree phylogenies and two supermatrix phylogenies. Based on the data and methods used, the authors suggest that the commonly used COI marker is not necessarily the best marker for reconstructing phylogenetic relationships or estimating population parameters in Mirid datasets.

Experimental design

To achieve their goals the authors sequenced 5 new mirid mitochondrial genomes and additionally retrieved 10 mt genomes from previous publications to compare them against. Several features of the alignments and genes (including the Ka/Ks ratio) were retrieved using MEGA, a popular suite of several phylogenetic methods and functions. For the phylogenetic analyses of their samples they defined the best partitioning scheme using PartitionFinder and consequently MrBayes and RAxML for reconstructing the phylogenetic relationships.

Validity of the findings

There are two important points that should be addressed with respect to the validity of the findings.

A) The reported values for the Ka/Ks ratio are alarmingly high, compared to what we normally expect for insect mitochondrial genomes (e.g., Meiklejohn et al., 2007, Basin et al., 2006). In general, the method and model used to estimate this ratio is of great importance as it may lead to substantial different estimations and errors (Angelis et al., 2014, Zhang and Yu, 2006). Given the unexpectedly high values recovered for the ω ratio in the present study it seems necessary that the authors provide further information and justification on the method and model they used and additionally infer the ratios using alternative models with proper statistical basis (e.g., bayesian methods, Angelis et al., 2014).

B) The authors discuss the in-congruence among the phylogenetic trees based on different mtDNA genes based on the recovered topology. I would encourage them to take bootstrap support values into account in their comparison (i.e. how many gene-topologies have high statistical support for the conflicting lineages?).

Additional comments

Minor comments

line 21 "The general mitogenomic features" please name them (or just a few) to make clearer what you mean

line 23 perhaps "the largest" fits better than "the most"

lines 53-54 ", largely due to lack of efficient molecular markers" This phrase doesn't make much sense.

line 56 "especially closely related species" something is missing here.

lines 66-67 how recently divergent were those species?

lines 81-83 rephrase to make clear where does the "were sequenced" phrase refer to.

lines 85-87 something is missing here.

line 89: It's not clear what does the "On the other hand" contradicts to.

line 111: Please specify here the source of the pulished sequences used.

line 129: it is not clear what the authors mean by "proof read the contigs", please explain or rephrase.

line 173: give the version of XSEDE here instead of line 181.

lines 174-175: add “each” after "...with four chains".

line 175: Replace "each set" with "each run".

lines 176-178: better "is assumed to be reached".

line 184: "Results and Discussion" change to "Results".

line 187: instead of "three were incomplete mitogenomes" you could report the percentage of completeness (e.g., one was sequences to 80% percent and two to 70%).

Results: the authors seem to have calculated the Ka/Ks ratio for all the coding genes and there are comments on these findings in "Results" (lines 258-284), however, I am not sure if these data are presented somewhere (I found the tables showing the Ka/Ks values for COI but not for other genes).

lines 192-194: redundant sentence.

lines 202: what is the "putative ancestral mitogenome"?

line 258/283 (and in other places of the manuscript): The term "evolutionary rate" should not be used with respect to the Ka/Ks ratio. Ka and Ks, independently, may be considered rates (even though not "evolutionary rates", however Ka/Ks is clearly a ratio rather than a rate).

lines 259-260: the phrase is confusing, try reforming it. For examples, "The largest genetic distances were found among subfamilies, followed by ... "

lines 261-263: provide some examples of such genes.

lines 266-268: instead of K2P value it should probably be K2P distance and “within” is probably better than "in".

line 279-280: "caused by high Ka and low Ks values" redundant.

line 313-314: "analytical analyses" ?

line 317 (319 similar): replace "other two" with "two other"

lines 341-342: "largely due to significant size difference between the two control regions." the same phrase was used earlier, repetitiveness makes the manuscript feel loose.

lines 342-345: the sentence is complicated and with repeated information, please rephrase it.

lines 377-379: it is not clear, please rephrase

lines 382-384: The genera cannot have Ka and Ks values, please rephrase to explain more clearly what you mean

line 388: something is missing here

lines 392-393: The authors should discuss here the most probable factors for retrieving incongruent phylogenies from different mtDNA loci. One of the most common reasons is the lack of adequate phylogenetic information within each individual gene, the authors should provide some insight to the relevance of this factor (i.e., are the conflicting lineages among gene-trees well supported?). However, they should further mention other potential factors such as incomplete taxon sampling (i.e., many missing related species).

line 409: "phylogenetic text"?


References

Meiklejohn, Colin D., Kristi L. Montooth, and David M. Rand. "Positive and negative selection on the mitochondrial genome." Trends in Genetics 23.6 (2007): 259-263.
Bazin, Eric, Sylvain Glémin, and Nicolas Galtier. "Population size does not influence mitochondrial genetic diversity in animals." Science 312.5773 (2006): 570-572.
Angelis, Konstantinos, Mario dos Reis, and Ziheng Yang. “Bayesian Estimation of Nonsynonymous/Synonymous Rate Ratios for Pairwise Sequence Comparisons.” Molecular Biology and Evolution 31.7 (2014): 1902–1913. PMC. Web. 14 May 2017.
Zhang, Zhang, and Jun Yu. "Evaluation of six methods for estimating synonymous and nonsynonymous substitution rates." Genomics, proteomics & bioinformatics 4.3 (2006): 173-181.

Reviewer 2 ·

Basic reporting

no comment (stated at the previous stage of reviewing)

Experimental design

no comment (stated at the previous stage of reviewing)

Validity of the findings

no comment (stated at the previous stage of reviewing)

Additional comments

I am very happy to acknowledge that Wang and colleagues have made a thorough revision of their manuscript in accordance to both reviewers comments and consider that their manuscript has been improved. Although they have made many changes both in terms of analysis (e.g. further investigation on cox-1 gene) and structure of the manuscript (e.g. the separation of Results, and Discussion sections, English language ms revision), the manuscript needs a minor revision, prior to its publication to PeerJ.
Although the authors provide additional information on the selection of the species under study they still need to elaborate a bit more. In the ms it is claimed that evolutionary rates of mitochondrial DNA genes have been evaluated. However, this is not accurate since only K2P distances have been estimated without any reference to divergence times. Please see below for a more detailed comment for this issue. Finally, at the Discussion section, text could be improved in order to have a better flow. Furthermore, there are some parts that belong either to the Materials and Methods or to the Results sections.
It could possibly be my mistake but I cannot seem to find the legends for Figures 1 to 3.
I also have some comments and suggestions for particular lines in the ms:
Abstract:
I believe that the abstract could be improved in order to reflect more accurately the aims, results and conclusions of the present study.
Please explain what an “ideal marker” would serve for in your case
Introduction:
L54-56 does that mean that the markers that have been employed are not informative enough? If yes can you please provide relevant references?
L82-84 when saying “all mirid mitogenomes” do you mean genomes sequences in the present study?
Materials & Methods
L135-136 please explain where/how did you get those mitogenomes
Results:
L186 Results (“and discussion” must be deleted)
L260-261 this statement is not accurate. In this case, evolutionary rates can be evaluated given that divergence time is known and cannot be evaluated through genetic distances alone. Please make the appropriate corrections throughout the ms.
Discussion:
L358 please replace “except” with “exception”
L359 as “in” other…
L363 probably instead of “intra-specific” you might want to consider “genus-specific”?
L366 please define “abnormal” i.e. be more specific
L379-384 this part belongs to the Materials & Methods section
L384-387 this part belongs to the Results section
L389 please add as well as other markers e.g. microsatellites and SNPs

---

## Round 0.3 · accepted · Accept

Thank you very much for the thoughtful revision of the manuscript. In the current form, your paper nicely refines the tools available for mirid bugs phylogenetic analysis and encourages researchers working with other taxa to re-evaluate the markers they use.